# Applying Evolutionary Multitasking for Process Parameter Optimization in Polymerization Process of Carbon Fiber Production

Liang Jin [1], Zude Zhou [1], Kunlun Li [2,*] , Guoliang Zhang [3], Quan Liu [2], Bitao Yao [1] and Yilin Fang [2]

1  School of Mechanical and Electronic Engineering, Wuhan University of Technology, 122 Luoshi Road, Wuhan 430070, China
2  School of Information Engineering, Wuhan University of Technology, 122 Luoshi Road, Wuhan 430070, China
3  Zhongfu Shenying Carbon Fiber Co., Ltd., No. 1-6 Jinqiao Road, Dapu Industrial Zone, Economic Development Zone, Lianyungang 222069, China
*  Correspondence: kunlunli@whut.edu.cn

**Abstract:** Carbon fiber is becoming a key material for engineering applications due to its excellent comprehensive properties. The process parameter optimization is an important step in the polymerization process of carbon fiber production. At present, most of the research on process parameter optimization is usually carried out on a single production line, without considering the correlation between optimization problems. In this paper, a multiobjective mechanism model for the co-optimization of the polymerization process of carbon fiber production is established. Each of these submodels is a multiobjective process parameter optimization task, corresponding to the polymerization process of a production line. In order to solve the model effectively, we also designed an evolutionary multitasking algorithm based on transfer learning, which reuses the past experiences of one task to generate a population pool for the next iteration of another task, enabling explicit genetic transfer between different tasks and accelerating the population convergence speed. The proposed multitasking framework for operation optimization has been conducted on 10 different production conditions of the polymerization process. Experimental results show that compared with other implicit and explicit genetic algorithms, this algorithm is very competitive in generating effective solutions. This research provides important support for process parameter optimization and manufacturing of carbon fiber production, which will help engineers and technicians to make informed decisions.

**Keywords:** polymerization process; carbon fiber production; process parameter optimization; co-optimization; evolutionary multitasking algorithm; transfer learning

## 1. Introduction

Polyacrylonitrile (PAN) and mesophase pitch (MP) are two of the most important precursors in the carbon fiber industry. The structure and composition of the precursor have an important influence on the performance of the carbon fiber obtained. Currently, the carbon fiber market is mainly dominated by PAN. Acrylonitrile (AN) polymerization is one of the key technologies in carbon fiber polymerization, and plays an important role in the preparation of high-performance carbon fiber [1–3]. The initiator decomposes at a certain temperature to produce free radicals, which causes initiate monomer chain polymerization, and accelerates the polymerization rate.

Figure 1 shows the carbon fiber production process. Carbon fiber is prepared from raw PAN precursor, which usually undergoes four processes: namely polymerization process, spinning process, oxidation stabilization process, carbonization and graphitization process. First, polymerization is the process of producing high-quality spinning stock solution to obtain high-quality copolymer by using precursors, and is the key to producing

high-performance carbon fiber [4,5]. Next, the purpose of the spinning process is to increase the speed of the cyclization reaction as well as to improve the orientation of the molecular chains in the fiber. Next, the oxidation stabilization process crosslinks the PAN chains in order to produce a molecular structure that is resistant to high temperature processing. Finally, the carbonization and graphitization process removes some carbon elements by subjecting the fiber to high-temperature processing to produce a high-strength carbon fiber with a stable structure. Generally, to improve the efficiency of industrial production, parallel process are used to provide good preparation for the subsequent process. In this paper, a co-optimization method of process parameters is proposed to solve the problem of multiple production lines during polymerization.

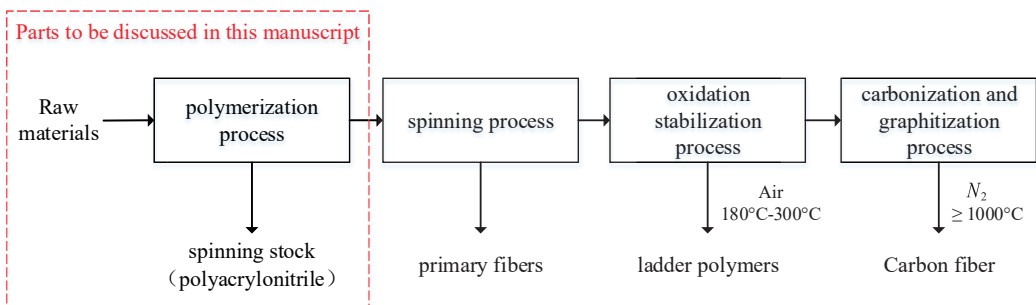

**Figure 1.** Carbon fiber production flow chart.

It is difficult to optimize process parameters of the polymerization process of carbon fiber production (PPCFP), mainly because of the following reasons. First, the PPCFP is complex, environmentally demanding and involves many processes and each of which has a significant impact on the quality of subsequent carbon fiber [6–8]. Optimization objectives are often contradictory and require some trade-offs. For example, resource efficiency and economic benefit are the two important production indices of PPCFP [9]. The more efficient the resource efficiency benefits, the lower the economic benefits. Therefore, PPCFP can be regarded as a typical multiobjective optimization problem (MOP). Finally, each production line (PL) in the parallel production process is very similar at the problem level. How to use the similarity of different tasks to optimize operational indices has become one of the important research issues in PPCFP.

In recent years, the operational indices of MOP in industrial production have extensively attracted attention. Yang et al. [10] fitted three kinds of assistant models via a large amount of historical production data in the beneficiation processes. The most accurate model is the optimization function in the actual production process, and the remaining model alternately assists the most accurate model for optimization. Taking into account five indicators such as iron concentrate output and concentrate grade for mineral processing production planning, Yu et al. [11] established a nonlinear multiobjective programming model. The author designed G-NSGA-II and G-SPEA2 based on a gradient hybrid operator to solve the model. Wang et al. [9] suggested integrating multiple mutation operators and adaptive selection strategy into an evolution algorithm (EA), which can effectively obtain better control parameters from the model in a continuous annealing production line model. Qian et al. [12] developed a multifidelity sequential optimization approach to determine the optimal design of the metamaterial vibration isolator. In the actual industrial optimization process, with the increase of the data dimensionality and the expansion of search space, the performance of the optimization algorithm is challenged. Moreover [13,14] put forward two search space reduction strategy algorithms, which provide a good idea to solve the above problems.

Inspired by Darwin's theory survival of the fittest [15], EAs mimic the natural extinction of organisms in nature, and can learn from individual or entire populations. EAs have also shown excellent performance in solving complex optimization problems such as MOPs in the real world, thanks to parallel processing, strong search ability and a wide range of

applications [16]. As one of the most active research fields in evolutionary optimization, multiobjective optimization EAs (MOEAs) have successfully been used in solving MOPs, which aims to strike a good balance between convergence and diversity [17,18]. At the same time, multitasking optimization has also attracted a lot of attention. It can handle more than one task at a time using a population, and each task can be one or more predefined objective functions. At this time, the optimization problem is classified into multifactorial optimization (MFO) [19,20]. With the help of the implicit parallelism search in the population and the related knowledge between tasks, the proposed multifactorial evolutionary algorithms (MFEAs) have successfully been used in solving MFO [21]. MFEAs have been successfully applied to many practical problems, such as job shop scheduling problems and expensive optimization problems [22,23].

Multipopulation approaches were developed in [24–26], where each subpopulation solved a task by exchanging genetic material between the multiple populations. For a promising performance, three strategies were adopted: Separated Genetic Algorithms, Bat Algorithms, and Variable Neighborhood Search. Zheng et al. [27] adopted an effective strategy based on random replacement via differential evolution to conduct knowledge transfer between populations, so that each subpopulation can efficiently handle the corresponding task. Feng et al. [28] proposed an Evolutionary Multitasking algorithm with explicit genetic transfer (EMT-EGT) that used a denoising autoencoder as a key component for inter-task knowledge transfer. By integrating the advantages of different solvers, the algorithm can deal with infeasible solutions effectively and avoid the algorithm converging to the local optimal solution. The resource allocation in the multitasking optimization was considered in [29]. The author believes that during population execution, acceptable solutions can be obtained via an online dynamic resource allocation strategy. Goh et al. [30] adopted hybridized competitive and cooperative mechanisms to optimize dynamic multi-objective problems. The MOP is divided into several subcomponents, each particular subcomponent of which will be completed by each subpopulation, and the eventual winners of the competition will cooperate with each other to guide the evolution process according to different optimization requirements. Yu et al. [31] developed an approach that was able to address these challenges through adaptive genetic and differential operators, a Gaussian mutation operator, and a memory-like strategy. Then, the optimal population location was gained effectively, which can help the convergence speed of the algorithm to accelerate. In Azzouz et al. [32] a dynamic constrained NSGA-II was developed using a more elaborated and self-adaptive penalty mechanism. Considering the characteristics of these methods, test benchmarks were designed. The algorithm maintains the advantages of convergence and diversity, can deal with infeasible solutions effectively and avoids the algorithm converging to the local optimal solution.

In this paper, we demonstrate an ideal multitasking optimization framework to solve the complex PPCFP. The framework takes full advantage of knowledge transfer across tasks for PPCFP. The main contributions of this article can be summarized as follows.

(1) Considering the correlation and characteristics of PPCFP, we established a multiobjective mechanism model for the co-optimization of the polymerization process. Each submodel is a multiobjective process parameter optimization task for a single production line of the polymerization process.

(2) EMT-EGT is based on different solvers, in which search mechanisms with the multiple solvers can be considered to accelerate the convergence speed between tasks. With this in mind, we proposed the transfer learning-based EGT (Tr-EGT), in which the past experience of one task is reused to generate the population pool for the next iteration of another task. Experimental results indicate that Tr-EGT achieves its effectiveness in addressing the PPCFP.

The rest of this paper is arranged as follows. In Section 2 we present the problem description and model for the polymerization process. In Section 3 we describe the details of the proposed algorithm. In Section 4 we will present the experimental results in incorpo-

rating our algorithm to solve the proposed model. In Section 5 we will draw a summary of this paper and outline the future research directions.

## 2. PPCFP Problem Formulation

We first review the PPCFP to be solved in this Section. On this basis, the mechanism model of the multiobjective process parameter optimization function is described.

### 2.1. Description of Polymerization Process

Figure 2 shows the general diagram of the polymerization unit.

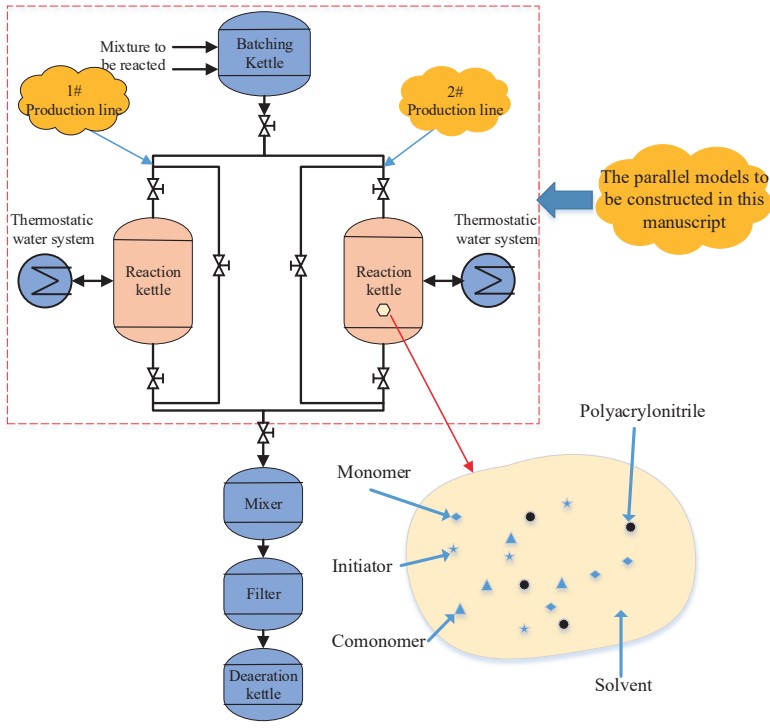

**Figure 2.** Schematic diagram of polymerization unit.

Generally speaking, the polymerization process can be divided into five stages: raw material preparation stage, polymerization reaction stage, demonomer stage, filtration stage, and defoaming stage. The main raw material for the preparation of carbon fibers is AN, which has the molecular formula $C_3H_3N$, contains side cyanide and is suitable for free radical polymerization. A common comonomer is itaconic acid (IA), which has the molecular formula $C_5H_6O_4$ and is introduced to improve the hydrophilicity of polyacrylonitrile by making the exothermic process more moderate and easily controlled. AN, comonomer, initiator, additives, and solvents enter into the reaction kettle, and a series of complex reactions occur under the action of the initiator. The chemical reaction equation is shown in Equation (1) [4].

$$n_1\, C_3H_3N + n_2\, C_5H_6O_4 + \text{additives} \xrightarrow{\text{Initiator, Solvent}} [C_3H_3N]n \qquad (1)$$

They blend completely at a certain temperature through the action of a stirrer. The polymerization of AN is usually triggered by free radicals, which releases reaction heat. Therefore, one of the key factors to ensure the stability of polymer quality is to take out the released reaction heat in time to achieve the dynamic balance of heat within the polymerizer. During the reaction, the rotation speed of the stirrer should be strictly controlled so that the molecular weight of the PAN can be stabilized in a certain range. Filters can remove gels and impurities from the polymer, allowing it to be pre-purified, thereby improving product quality. Under certain temperature and vacuum conditions, the bubbles in the polymer

solution are removed, and the PAN raw liquid is transported to the spinning stage after the defoaming.

### 2.2. Optimization Modeling

In this paper, the correlation between optimization problems is considered, and process parameter optimization is studied via a multiobjective multifactorial mechanism model of PPCFP. In the mechanism model, the yield (Q) and the production capacity utilization (C) are taken as objections in a different production lines. We use operational indices as decision variables and constraints.

#### 2.2.1. Parameters and Variables

The aim of process parameter optimization is to improve the global production efficiency of PPCFP through the synergistic mechanism model. The model consists of two optimization tasks of the polymerization process of a production line, in which each submodel deals with a single task. The symbols and meanings of the model used in our formulation are described in Table 1.

**Table 1.** Symbols and meanings.

| Symbol | Meaning | Symbol | Meaning |
|---|---|---|---|
| $Q_{1,1}$ | Yield in PL 1# | $w_1$ | Weight factor |
| $Q_{2,1}$ | Yield in PL 2# | $w_2$ | Weight factor |
| $C_{1,2}$ | production capacity utilization in PL 1# | $w_3$ | Weight factor |
| $C_{2,2}$ | production capacity utilization in PL 2# | $k$ | Tradeoff factor |
| $M_1$ | Monomer concentration of kettle 1 | $T_s$ | Temperature of material mass flow |
| $M_2$ | Monomer concentration of kettle 2 | $q_{M_1}$ | Material mass flow of kettle 1 |
| $I_1$ | Initiator concentration of kettle 1 | $q_{M_2}$ | Material mass flow of kettle 2 |
| $I_2$ | Initiator concentration of kettle 2 | $H$ | Specific heat capacity |
| $N_1$ | Comonomer concentration of kettle 1 | $t$ | Production time |
| $N_2$ | Comonomer concentration of kettle 2 | $\beta$ | Correction parameters |
| $T_1$ | Temperature of kettle 1 | $m$ | Material quality |
| $T_2$ | Temperature of kettle 2 | $R$ | Idea gas constant |
| $M_{min}$ | Lower bound of monomer concentration | $U_{tra}$ | Heat transfer coefficient |
| $M_{max}$ | Upper bound of monomer concentration | $A_{tra}$ | Heat exchange contact area |
| $I_{min}$ | Lower bound of initiator concentration | $\overline{T_{jac}}$ | Jacket average temperature |
| $I_{max}$ | Upper bound of initiator concentration | $k_p$ | Chain growth rate constant |
| $N_{min}$ | Lower bound of comonomer concentration | $f$ | Initiation efficiency |
| $N_{max}$ | Upper bound of comonomer concentration | $k_t$ | Material mass flow |
| $k_{d1}$ | Decomposition rate constant of kettle 1 | $A_d$ | Pre-exponential factor |
| $k_{d2}$ | Decomposition rate constant of kettle 2 | $E_d$ | Activation energy |

In order to ensure the safety and reliability of the polymerization process, every decision variable needs to satisfy a certain range in the actual industrial production. Table 2 shows the fixed parameters and their values. The ranges of every decision variable are presented in Table 3.

**Table 2.** Fixed parameters and their values.

| Parameters | Value | Parameters | Value |
|---|---|---|---|
| $w_1$ | 0.49315 | $H$ | 4179 J/(kg·K) |
| $w_2$ | 1 | $\beta$ | 0.99 |
| $w_3$ | 1 | $f$ | 0.8 |
| $k$ | 1 | $R$ | 8.314 J/(mol·K) |
| $A_{tra}$ | 1 m$^2$ | $A_d$ | $1 \times 10^{13}$ |
| $\overline{T_{jac}}$ | 333.15 K | $E_d$ | 120 kJ/mol |
| $k_p$ | 1960 L/(mol·S) | $k_t$ | $7.82 \times 10^8$ L/(mol·S) |

**Table 3.** Boundaries of decision variables.

| Name | $M_1$ | $I_1$ | $N_1$ | $M_2$ | $I_2$ | $N_2$ |
|---|---|---|---|---|---|---|
| Lower bound | 0.1 | 0.01 | 0.01 | 0.1 | 0.01 | 0.01 |
| upper bound | 1 | 0.1 | 0.1 | 1 | 0.1 | 0.1 |
| Unit | mol/L | mol/L | mol/L | mol/L | mol/L | mol/L |

### 2.2.2. Production Conditions

Production conditions often change due to the properties of polymerization process. To ensure that the production index is in the most optimum condition under different production conditions, we need to re-optimize process parameters of any given production condition. Taking this cue, we investigated the optimization of 10 different production conditions during the polymerization process. These production conditions are set out in Table 4.

**Table 4.** 10 typical operational conditions in production process.

| Name | t(S) | $T_1$(T) | $q_{M_1}$(g/S) | $T_2$(T) | $q_{M_2}$(g/S) |
|---|---|---|---|---|---|
| Conditon 1 | 1 | 333.96 | 1054 | 333.97 | 946 |
| Conditon 2 | 2 | 333.77 | 1028 | 333.78 | 972 |
| Conditon 3 | 3 | 333.57 | 1042 | 333.62 | 958 |
| Conditon 4 | 4 | 333.35 | 1084 | 333.47 | 916 |
| Conditon 5 | 5 | 333.23 | 1001 | 333.23 | 999 |
| Conditon 6 | 6 | 333.04 | 1012 | 333.06 | 988 |
| Conditon 7 | 7 | 332.78 | 1067 | 332.95 | 933 |
| Conditon 8 | 8 | 332.57 | 1083 | 332.80 | 917 |
| Conditon 9 | 9 | 332.48 | 1014 | 332.53 | 986 |
| Conditon 10 | 10 | 332.22 | 1058 | 332.42 | 942 |

### 2.2.3. Modeling

From the above analysis, in order to optimize both the yield (Q) and the production capacity utilization (C) of two tasks simultaneously, optimization objectives can be defined as follows [33,34].

$$Task1: \begin{cases} Maximize \quad Q_{1,1} = \\ \quad k_p(\dfrac{f*k_{d1}}{k_t})^{\frac{1}{2}}(I_1*e^{-\frac{t}{10}})^{\frac{1}{2}}M_1e^{-10N_1} \quad (2) \\ Maximize \quad C_{1,2} = \\ \quad \dfrac{k}{e^{(w_1*M_1+w_2*I_1+w_3*N_1)}e^{-(334.15-T_1)}} \quad (3) \end{cases}$$

$$Task2: \begin{cases} Maximize \quad Q_{2,1} = \\ \quad k_p(\dfrac{f*k_{d2}}{k_t})^{\frac{1}{2}}(I_2*e^{-\frac{t}{10}})^{\frac{1}{2}}M_2e^{-10N_2} \quad (4) \\ Maximize \quad C_{2,2} = \\ \quad \dfrac{k}{e^{(w_1*M_2+w_2*I_2+w_3*N_2)}e^{-(334.15-T_2)}} \quad (5) \end{cases}$$

$$s.t. \begin{cases} M_{min} \leq M_i \leq M_{max}, \quad i=1,2. \quad (6) \\ I_{min} \leq I_i \leq I_{max}, \quad\quad i=1,2. \quad (7) \\ N_{min} \leq N_i \leq N_{max}, \quad i=1,2. \quad (8) \end{cases}$$

As the Equations (3) and (5) show, the $T_i$ require to be calculated by mathematical formulas. According to the heat conservation theorem, the change of heat in the polymerization reactor is equal to the sum of the input and output energy of the system. For simplicity,

the changes in chemical reaction heat, kinetic energy, and potential energy of solution in the polymerization reactor are negligible. The formula is presented in Equation (9) [35].

$$mH\frac{dT_i}{dt} = q_{M_i}H(T_{si} - T_i)$$
$$+ U_{tra}A_{tra}(\overline{T_{jac}} - T_i), i = 1, 2. \tag{9}$$

Assume the temperature of the solution $T_s$ during the time from the polymer reservoir to the polymerizer satisfies exponential distribution. The formulas are shown in Equation (10).

$$T_{si} = \begin{cases} \mu_i e^{-\lambda_i t} & \text{if } t \geq 0, i = 1, 2. \\ 0 & \text{others} \end{cases} \tag{10}$$

As shown in Equations (2) and (4), the $k_d$ requires to be calculated, which is likely to vary with ambient temperature [36], and its relationship with temperature can be expressed by the widely used Arrhenius formula. The formula is presented in Equation (11).

$$k_{di} = A_d e^{-(\frac{E_d}{RT_i})^\beta}, i = 1, 2. \tag{11}$$

## 3. Proposed Algorithm

This section details the multitasking optimization framework of the proposed Tr-EGT. The offspring generation of Tr-EGT using TCA is also introduced.

### 3.1. Framework of Tr-EGT

The flowchart of the proposed Tr-EGT algorithm for operational indices optimizations of the polymerization process is shown in Figure 3.

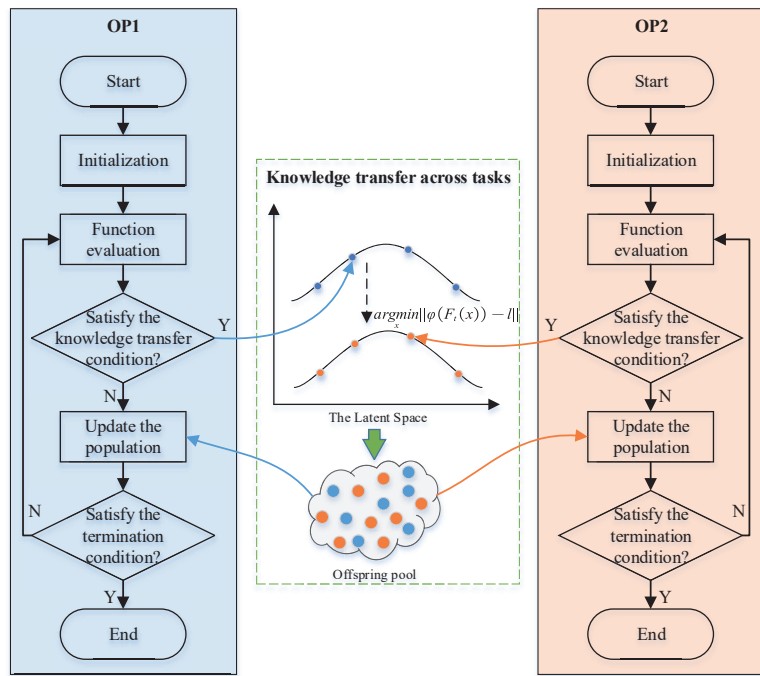

**Figure 3.** Flow chart of the proposed Tr-EGT.

First, for each optimization problem, i.e., $OP_1$ and $OP_2$, we assign it a single-task evolutionary solver with an independent population. Next, to establish the connections between different optimization problems, two sets of solutions are randomly sampled from the corresponding search space as the source domain and target domain of the

domain adaptation approach, and a transformation mapping matrix across tasks is obtained. Further, the explicit genetic transfer occurs when the two evolutionary solvers performed at a certain interval. In particular, during the execution of both solvers, a good solution sought by one solver is injected into the population of the other through a domain adaptation approach.

Algorithm 1 demonstrates the framework of Tr-EGTto help the readers better understand the flowchart.

---

**Algorithm 1** Framework of Tr-EGT

---

**Require:** $G_{max}$, maximum generation; $G$, interval of explicit solution transfer across tasks; $F_1(.)$, the optimization function for $OP_1$; $F_2(.)$, the optimization function for $OP_2$; solver1, a single-task evolutionary solver for $OP_1$; solver2, a single-task evolutionary solver for $OP_2$; $\kappa(\cdot, \cdot)$, kernel function.

**Ensure:** the $POF_s$ of each task.
1:  Initialization two populations and set g = 0.
2:  Calculate $F_{OP_1}$ and $F_{OP_2}$ to get the $POF1_g$ and $POF2_g$, respectively via use solver1 and solver2.
3:  **for** g = 1 to $G_{max}$ **do**
4:      **if** mod (g, G) = 0 **then**
5:          Next-Offspring$_1$ =Tr-OG($F_1(.)$, $POF2_g$, $\kappa(\cdot, \cdot)$).
6:          Next-Offspring$_2$ =Tr-OG($F_2(.)$, $POF1_g$, $\kappa(\cdot, \cdot)$).
7:      **else**
8:          Generate Next-Offspring$_1$ and Next-Offspring$_2$, respectively using assortative mating.
9:      **end if**
10:     $POP1_{g+1} = POP1_g \cup \{$Next-Offspring$_1\}$.
11:     $POP2_{g+1} = POP2_g \cup \{$Next-Offspring$_2\}$.
12:     Update the $POP1_{g+1}$ and $POP2_{g+1}$.
13:     $POF1_{g+1}$=solver1($POP1_{g+1}$).
14:     $POF2_{g+1}$=solver2($POP2_{g+1}$).
15:     g = g + 1.
16: **end for**

---

Tr-EGT follows the general framework of EMT-EGT, except that the transfer learning-based offspring generator (Tr-OG) is adopted to conduct efficient knowledge transfer across tasks. $F_1$ and $F_2$ are the optimization functions with respect to $OP_1$ and $OP_2$, respectively, and $POF1_g$ and $POF2_g$ represent their POF, which has already been found. Next-Offspring$_1$ and Next-Offspring$_2$ are the offspring for $F_1$ and $F_2$ at the next generation. When the interval conditions are met, Tr-OG generates an offspring population using the POF of domain adaption approach. A new generation is chosen from the joint of the offspring population and its parent population via environment selection, which can be used to evolve at generation $g + 1$. Specifically, we can get two types of knowledge transfer; in general, one is $OP_1 \rightarrow OP_2$, and the other is $OP_2 \rightarrow OP_1$. For $OP_1 \rightarrow OP_2$, the obtained $POF1_g$ and $F_2(.)$. In general, the feasible solutions of $OP_2$, are used as the source domain and target domain, respectively. A mapping function $\varphi$ is got using the domain adaptation approach, which maps source and target data with different distributions into the latent space. Next, a good offspring population is generated, which accelerates convergence of the population.

### 3.2. Offspring Generation Using TCA

In recent years, more and more attention has been paid to transfer learning, which can effectively solve issues between different problems by conducting knowledge transfer [37,38]. Domain adaption is an important branch of transfer learning and transfer component analysis (TCA) is commonly used to reuse the knowledge across problems [39,40]. In this paper, TCA is applied to EGT to minimize the two different probability distributions

for source and target domains in latent space [41]. In the reproducing kernel Hilbert space (RKHS), the distance between two different distributions can be calculated using a kernel function. Based on the RKHS, the maximum mean discrepancy (MMD) is employed to minimize the discrepancy between two distributions [41]. To retain the key properties and reduce computational complexity, we construct the kernel function $\kappa$ from the data to obtain a low-rank transformation matrix $W$.

The TCA algorithm for solving optimization problems can be presented as follows.

$$\arg\min_{W} \mu \cdot \text{tr}\left(W^T W\right) + \text{tr}\left(W^T KLKW\right)$$

$$\text{subject to } W^T KHKW = \mathbf{I}$$

(12)

where $\mathbf{H} = \mathbf{I} - (1/[m+n])\mathbf{1}\mathbf{1}^T$ and $\mathbf{I} \in \mathbf{R}^{(m \times n) \times (m \times n)}$. $m$ and $n$ denote the numbers of sampled solutions for source and target data. The solution of Equation (12) can be solved by the well-known generalized eigenvalue decomposition [42].

In the following algorithm, Tr-OG algorithm (provide pseudo code in Algorithm 2), taking sampled solutions in $F_s(X_s)$ and $F_t(Y_t)$ as the input to the TCA, the transformation matrix $W$ can be naturally obtained. The latent space (LS) is constructed with the $W$. Further, a decision variable $x$ is searched in LS according to $\|\varphi(F_t(x)) - l\|$ (line 6), which is used to generate the offspring population for $F_t(Y_t)$.

---

**Algorithm 2** Framework of Tr-OG

---

**Require:** $F_s(.)$, the optimization function of source domain; $F_t(.)$, the optimization function of target domain; $POF_g$, the POF of the function $F_s(.)$; $\kappa(\cdot, \cdot)$, kernel function.
**Ensure:** A offspring for $F_t(.)$.
　1: Initialization.
　2: Randomly sample two sets of the solutions with respect to $F_s(.)$ and $F_t(.)$, defined by $X_s$ and $Y_t$, respectively.
　3: Calculate $F_s(X_s)$ and $F_t(Y_t)$ of two different optimization problems.
　4: Obtain a transformation matrix W through TCA $\{F_s(X_s), F_t(Y_t), \kappa(\cdot, \cdot)\}$.
　5: Construct the LS by using a mapping function $\varphi(.)$.
　6: Generate a offspring for $F_t(.)$ via $\arg\min_{x}\|\varphi(F_t(x)) - l\|, l \in$ LS.

---

For example, the Tr-EGT algorithm is used to solve the optimization problem $OP_1$, which contains two objectives of yield and production capacity utilization. Suppose we have obtained the POF of $OP_1$ problem, $POF_g$ and $p \in POF_g$. A mapping function $\varphi(.)$ is obtained via the TCA algorithm, which is used to map the $p$ into a high-dimension LS. Then a solution $x$ for the $OP_2$ problem at generation g+1 will be found by using the Tr-EGT algorithm, and this solution x meets the requirements which is closest to $l$ in the LS. This solution $x$ will be output as one of the offspring with regard to the initial population, which can help the $OP_2$ problem at generation g+1 solve faster. At the same time, when we get the POF of the $OP_2$ problem, the above method can also speed up the solution of $OP_1$ faster, thereby achieving an efficient solution of the problem between the two tasks.

## 4. Experimental Studies

In order to verify the validity of the proposed multitask algorithmic framework, we present a case study of 10 different operating conditions of the polymerization process in this section.

In this experiment, hypervolume (HV) [43,44] is used as a performance metric to compare results between different algorithms. A large HV value means that the algorithm has better diversity and convergence. Accuracy in calculating the HV index depends on the selection of reference points. First, We calculate the maximum value of the boundary in different environments, multiply the maximum value by 1.1 as the reference point for different environments, and then the reference point in different environments is selected

for evaluation. The solved HV value is then normalized. Next, we use the normalized HV metric to compare the diversity and convergence of different algorithms.

### 4.1. Parameter Settings

Following [28], two types of solvers are required to optimize two tasks, i.e., one is SPEA2, and the other is NSGA-II. The interval for explicit knowledge transfer is set to ten. The *rmp* of MO-MFEA is set to 0.3. To fairly compare the performance of algorithms, the population size of MO-MFEA, EMT-EGT, and Tr-EGT is set to 200, and the single task-based NSGA-II population scale is set to 100. When the maximum generation of 200 is reached, all algorithms are terminated and its independent number of runs is 20. Table 5 shows the parameter details of all algorithms.

**Table 5.** Parameter setting for the multitasking multiobjective experiment.

| Algorithm | rmp | SBX $\eta_c$ | PM $\eta_m$ | N | G | Maxgen | Runs |
|-----------|-----|--------------|-------------|-----|-----|--------|------|
| NSGA-II | - | 20 | 20 | 100 | - | 200 | 20 |
| MOMFEA | 0.3 | 20 | 20 | 200 | - | 200 | 20 |
| EMT-EGT | - | 20 | 20 | 200 | 10 | 200 | 20 |
| Tr-EGT | - | 20 | 20 | 200 | 5 | 200 | 20 |

### 4.2. Simulation Results and Discussion

The experimental results of these four algorithms, i.e., NSGA-II, MO-MFEA, EMT-EGT, and Tr-EGT, on two tasks under 10 operational conditions are listed in Tables 6 and 7, respectively.

The average HV values over 20 independent runs of all algorithms are shown, and we highlight the best result for each task in boldface. The best run result for each test case is highlighted. The Wilcoxon rank sum test at the significance level of 0.05 is employed, where the symbol "$+ / - / \approx$" indicates that the result of the corresponding algorithm is significantly better, significantly worse, and comparable to that Tr-EGT, respectively.

It can be observed that Tr-EGT performs better than NSGA-II, MO-MFEA, and EMT-EGT, which indicates that the strategy of transfer learning-based explicit genetic transfer can accelerate the convergence speed of the population. EMT-EGT only obtains the best results on T1 of 10 operational conditions using an evolutionary solver with NSGA-II, while NSGA-II performs well in all tasks, but may not take full advantage of different solvers (i.e., NSGA-II and SPEA2), leading in the occurrence of negative transfer. From the MO-MFEA experiments, we can see that it does not always help to solve a strong correlation e.g., PPCFP model, and harnessing the inductive bias may hurt, thereby leading to the possibility of negative knowledge transfer.

To visualize the performance of all algorithms during execution, Figure 4 shows the average HV numerical curves over 20 independent runs on task$_1$ under all conditions.

**Table 6.** Averaged HV value and Standard Deviation obtained by NSGA-II, MO-MFEA, EMT-EGT, and Tr-EGT of $task_1$ on ten conditions.

| Problem | Task | NSGA-II | MO-MFEA | EMT-EGT | Tr-EGT |
|---|---|---|---|---|---|
| Conditon 1 | T1 | $2.656 \times 10^{-1}(3.036 \times 10^{-2})\approx$ | $2.269 \times 10^{-1}(2.399 \times 10^{-3})-$ | $2.733 \times 10^{-1}(1.938 \times 10^{-2})\approx$ | $\mathbf{2.753 \times 10^{-1}(1.668 \times 10^{-2})}$ |
| Conditon 2 | T1 | $2.907 \times 10^{-1}(2.818 \times 10^{-2})-$ | $2.562 \times 10^{-1}(3.114 \times 10^{-3})-$ | $3.006 \times 10^{-1}((2.533 \times 10^{-2})\approx$ | $\mathbf{3.120 \times 10^{-1}(1.992 \times 10^{-2})}$ |
| Conditon 3 | T1 | $3.446 \times 10^{-1}(3.406 \times 10^{-2})\approx$ | $2.939 \times 10^{-1}(2.706 \times 10^{-3})-$ | $3.601 \times 10^{-1}(3.100 \times 10^{-2})-$ | $\mathbf{3.624 \times 10^{-1}(2.664 \times 10^{-2})}$ |
| Conditon 4 | T1 | $3.619 \times 10^{-1}(3.322 \times 10^{-2})\approx$ | $3.056 \times 10^{-1}(2.758 \times 10^{-3})-$ | $3.537 \times 10^{-1}(3.135 \times 10^{-2})\approx$ | $\mathbf{3.761 \times 10^{-1}(2.651 \times 10^{-2})}$ |
| Conditon 5 | T1 | $3.359 \times 10^{-1}(2.600 \times 10^{-2})\approx$ | $2.792 \times 10^{-1}(2.805 \times 10^{-3})-$ | $3.237 \times 10^{-1}(2.839 \times 10^{-2})-$ | $\mathbf{3.482 \times 10^{-1}(3.380 \times 10^{-2})}$ |
| Conditon 6 | T1 | $4.363 \times 10^{-1}(2.892 \times 10^{-2})-$ | $3.730 \times 10^{-1}(1.648 \times 10^{-3})-$ | $4.428 \times 10^{-1}(4.244 \times 10^{-2})\approx$ | $\mathbf{4.608 \times 10^{-1}(2.833 \times 10^{-2})}$ |
| Conditon 7 | T1 | $4.660 \times 10^{-1}(4.394 \times 10^{-2})-$ | $4.034 \times 10^{-1}(4.132 \times 10^{-3})-$ | $4.702 \times 10^{-1}(4.309 \times 10^{-2})-$ | $\mathbf{5.067 \times 10^{-1}(4.459 \times 10^{-2})}$ |
| Conditon 8 | T1 | $4.432 \times 10^{-1}(3.857 \times 10^{-2})-$ | $3.761 \times 10^{-1}(3.068 \times 10^{-3})-$ | $4.478 \times 10^{-1}(3.214 \times 10^{-2})-$ | $\mathbf{4.808 \times 10^{-1}(3.380 \times 10^{-2})}$ |
| Conditon 9 | T1 | $3.759 \times 10^{-1}(3.875 \times 10^{-2})\approx$ | $3.227 \times 10^{-1}(3.379 \times 10^{-3})-$ | $3.808 \times 10^{-1}(2.878 \times 10^{-2})-$ | $\mathbf{3.981 \times 10^{-1}(2.673 \times 10^{-2})}$ |
| Conditon 10 | T1 | $5.036 \times 10^{-1}(5.163 \times 10^{-2})\approx$ | $4.196 \times 10^{-1}(3.861 \times 10^{-3})-$ | $4.979 \times 10^{-1}(4.177 \times 10^{-2})\approx$ | $\mathbf{5.067 \times 10^{-1}(2.843 \times 10^{-2})}$ |

**Table 7.** Averaged HV value and Standard Deviation obtained by NSGA-II, MO-MFEA, EMT-EGT, and Tr-EGT of $task_2$ on ten conditions.

| Problem | Task | NSGA-II | MO-MFEA | EMT-EGT | Tr-EGT |
|---|---|---|---|---|---|
| Conditon 1 | T2 | $2.657 \times 10^{-1}(2.602 \times 10^{-2})\approx$ | $2.229 \times 10^{-1}(2.337 \times 10^{-3})-$ | $2.251 \times 10^{-1}(3.685 \times 10^{-4})-$ | $\mathbf{2.678 \times 10^{-1}(2.138 \times 10^{-2})}$ |
| Conditon 2 | T2 | $3.074 \times 10^{-1}(2.742 \times 10^{-2})\approx$ | $2.518 \times 10^{-1}(3.080 \times 10^{-3})-$ | $2.545 \times 10^{-1}(5.238 \times 10^{-4})-$ | $\mathbf{3.189 \times 10^{-1}(2.547 \times 10^{-2})}$ |
| Conditon 3 | T2 | $3.285 \times 10^{-1}(3.384 \times 10^{-2})\approx$ | $2.821 \times 10^{-1}(2.634 \times 10^{-3})-$ | $2.843 \times 10^{-1}(5.461 \times 10^{-4})-$ | $\mathbf{3.437 \times 10^{-1}(3.352 \times 10^{-2})}$ |
| Conditon 4 | T2 | $3.246 \times 10^{-1}(3.580 \times 10^{-2})\approx$ | $2.733 \times 10^{-1}(2.296 \times 10^{-3})-$ | $2.757 \times 10^{-1}(5.343 \times 10^{-4})-$ | $\mathbf{3.387 \times 10^{-1}(3.145 \times 10^{-2})}$ |
| Conditon 5 | T2 | $3.301 \times 10^{-1}(3.049 \times 10^{-2})-$ | $2.799 \times 10^{-1}(2.790 \times 10^{-3})-$ | $2.824 \times 10^{-1}(4.715 \times 10^{-4})-$ | $\mathbf{3.590 \times 10^{-1}(2.667 \times 10^{-2})}$ |
| Conditon 6 | T2 | $4.290 \times 10^{-1}(4.162 \times 10^{-2})\approx$ | $3.647 \times 10^{-1}(1.724 \times 10^{-3})-$ | $3.676 \times 10^{-1}(6.411 \times 10^{-4})-$ | $\mathbf{4.348 \times 10^{-1}(3.135 \times 10^{-2})}$ |
| Conditon 7 | T2 | $4.000 \times 10^{-1}(3.476 \times 10^{-2})-$ | $3.446 \times 10^{-1}(3.429 \times 10^{-3})-$ | $3.487 \times 10^{-1}(6.301 \times 10^{-4})-$ | $\mathbf{4.184 \times 10^{-1}(2.688 \times 10^{-2})}$ |
| Conditon 8 | T2 | $3.524 \times 10^{-1}(3.959 \times 10^{-2})\approx$ | $3.018 \times 10^{-1}(2.604 \times 10^{-3})-$ | $3.042 \times 10^{-1}(4.283 \times 10^{-4})-$ | $\mathbf{3.640 \times 10^{-1}(2.765 \times 10^{-3})}$ |
| Conditon 9 | T2 | $3.808 \times 10^{-1}(4.134 \times 10^{-2})\approx$ | $3.098 \times 10^{-1}(3.264 \times 10^{-3})-$ | $3.128 \times 10^{-1}(6.248 \times 10^{-4})-$ | $\mathbf{3.863 \times 10^{-1}(2.706 \times 10^{-2})}$ |
| Conditon 10 | T2 | $4.122 \times 10^{-1}(3.911 \times 10^{-2})\approx$ | $3.460 \times 10^{-1}(3.048 \times 10^{-3})-$ | $3.493 \times 10^{-1}(4.789 \times 10^{-4})-$ | $\mathbf{4.269 \times 10^{-1}(2.758 \times 10^{-2})}$ |

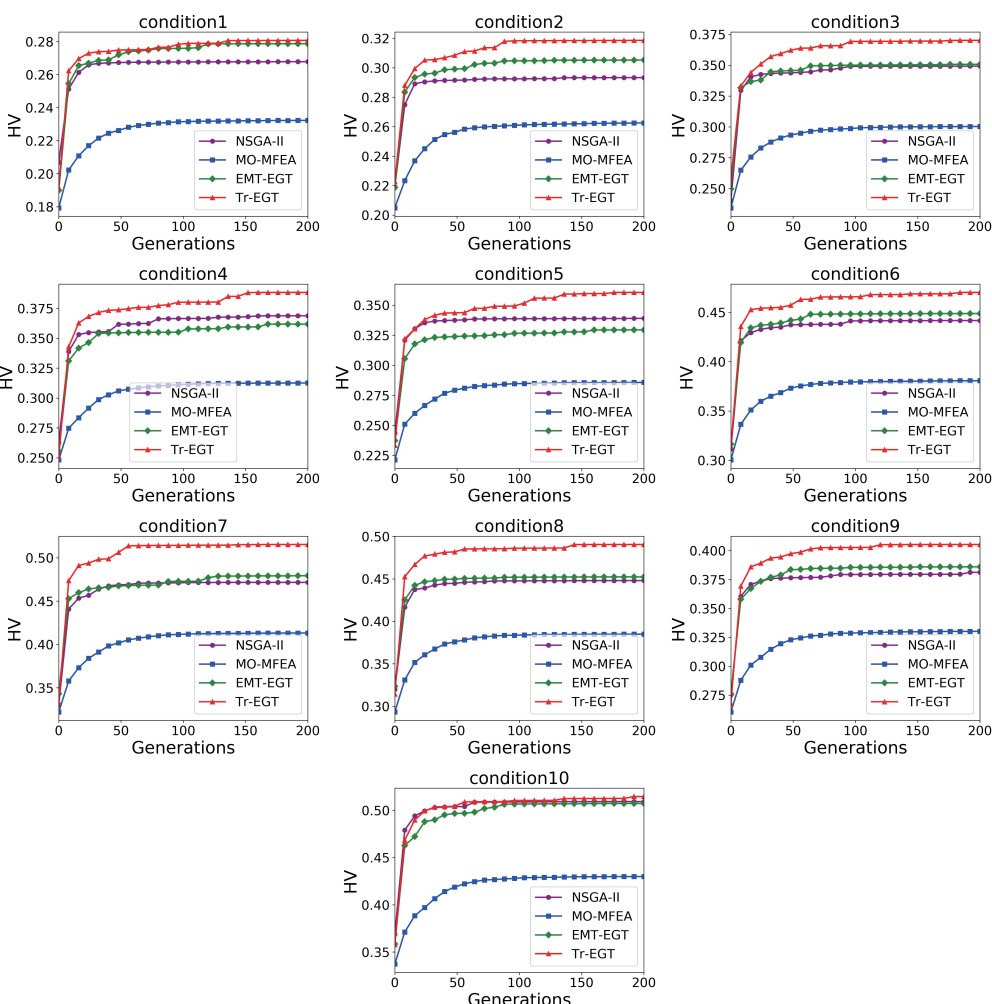

**Figure 4.** Average HV numerical curves of NSGA-II, MO-MFEA, EMT-EGT and Tr-EGT over 20 independent runs on $task_1$.

The MO-MFEA is generally stagnant and it is easy to get stuck in local optimal regions. Meanwhile, The EMT-EGT and NSGA-II algorithms achieve a good performance may be attributed to formulated PPCFP. It can be also observed that Tr-EGT converges rapidly under all conditions in the early optimization stages, which is attributed to the fact that it employs the Tr-OG strategy to autonomously exploit latent complementarities between two different problems, thereby successfully improving the positive knowledge transfer. Since the proposed Tr-EMT algorithm shares the same evolutionary solver as NSGA-II, the only difference is that the proposed method contains explicit genetic transfer between different tasks, the convergence speed of the average HV numerical curve confirms the effectiveness of performing Tr-EMT for optimization. Moreover, compared to the recently proposed benchmark algorithm MO-MFEA with implicit genetic transfer for multiobjective multitask optimization, the Tr-EMT algorithm obtains superior solution quality in terms of the average HV value across all tasks, which further confirms the effectiveness of the proposed Tr-EMT with explicit genetic transfer across tasks. Even if MO-MFEA and NSGA-II share the same evolutionary search operator, with the help of the implicit cross-task genetic transfer, in general, MOMFEA will show faster convergence than NSGA-II on multiobjective tasks. However, experimental results show the opposite, with NSGA-II having a faster convergence rate, which also suggests that negative transfer occurs even with the higher similarity between tasks.

The average HV values over 20 independent runs on $task_2$ under all conditions are shown in Figure 5. It can be observed that for the EMT-EGT algorithm, inductive bias from

another task fails to relieve premature convergence, resulting in deteriorated performance. Like Figure 4, the MO-MFEA also falls into a local optimal.

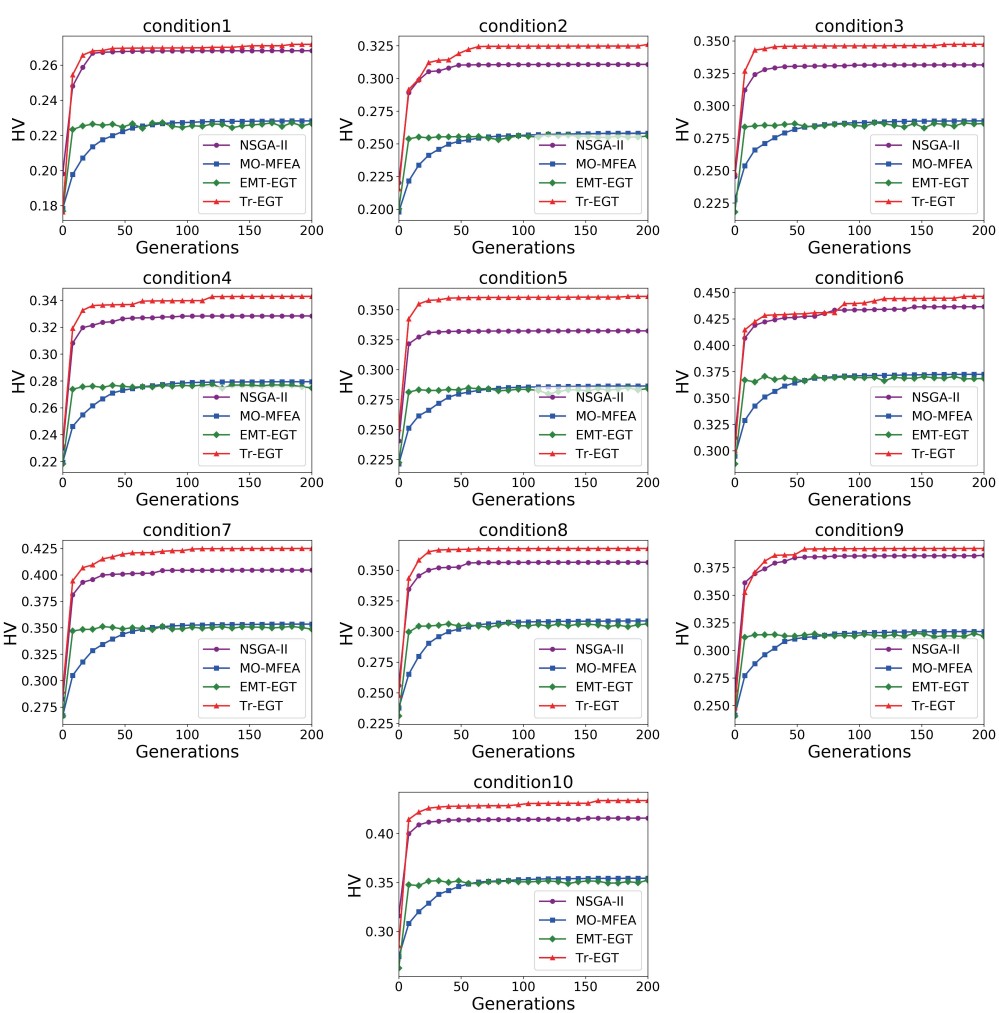

**Figure 5.** Average HV numerical curves of NSGA-II, MO-MFEA, EMT-EGT and Tr-EGT over 20 independent runs on task$_2$.

Table 8 shows the mean computational times of the four algorithms for tasks under conditions 1, 7, and 8. Among them, MO-MFEA, EMT-EGT, and EMT-EGT, as multitasking algorithms, are able to optimize tasks T1 and T2 at the same time, with values in the table being the sum of the time of both tasks. The Tr-EGT algorithm takes more time than EMT-EGT, NSGA-II and MO-MFEA. There are two main reasons for this. First, in the EMT-EGT algorithm, the MMD distance between the solution set of one task and the environment of another needs to be calculated to generate the initialized population pool for the iteration process of the next task, thus leading to accelerate the convergence speed of the algorithm. Secondly, we set a small value for inter-task knowledge transfer frequency G relative to Tr-EGT, which means an increase in the frequency of computing MMD distance. However, the additional computational time added in the EMT-EGT algorithm is acceptable for the optimization process in the carbon fiber industry.

**Table 8.** Mean computational times obtained between different algorithms for Conditions 1, 7, 8 over 20 runs in task₁ and task₂ (in seconds).

| Problem | Task | NSGA-II | MO-MFEA | EMT-EGT | Tr-EGT |
|---------|------|---------|---------|---------|--------|
| condition 1 | T1 | 18.19 | 36.23 | 70.26 | 404.89 |
| condition 1 | T2 | 17.86 | | | |
| condition 7 | T1 | 18.84 | 37.50 | 75.77 | 354.51 |
| condition 7 | T2 | 18.73 | | | |
| condition 8 | T1 | 19.30 | 38.89 | 76.15 | 345.98 |
| condition 8 | T2 | 18.75 | | | |

Figure 6 shows the average approximation PFs among 20 runs obtained by the MO-MFEA, EMT-EGT, and Tr-EGT in the objective space under conditions 1, 7, and 8. It can be observed that the solutions got by the Tr-EGT can be distributed to a large area. In other words, under these conditions, the Tr-EGT algorithm can find a high production capacity utilization with an acceptable yield. With this information, the Tr-EGT can achieve better diversity and convergence than the traditional multitasking framework, thus helping the engineers/technicians to make informed decisions.

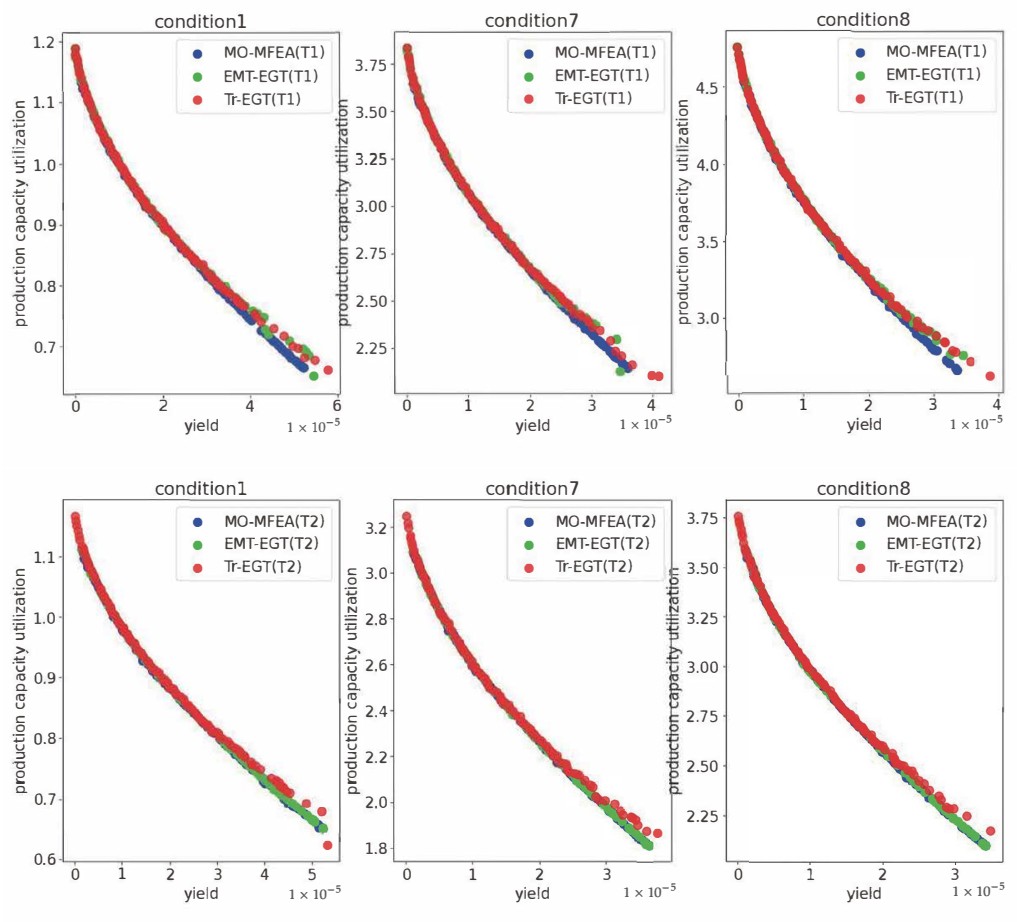

**Figure 6.** The average approximate PF was obtained by the multitask algorithms under Conditions 1, 7, and 8 over 20 runs in task₁ and task₂.

## 5. Conclusions and Future Work

This article investigates the process parameter optimization problem of PPCFP, and establishes a multiobjective multifactorial mechanism model. In the model, each submodel is a multiobjective process parameter optimization task. Next, a multitasking framework Tr-EGT was proposed to treat the PPCFP. The main difference between Tr-EGT and existing

EGT algorithms lies in the Tr-OG strategy. The Tr-OG strategy establishes the connection across tasks via a transformation matrix, which employs the transfer learning approach to generate an offspring population, accelerating the convergence of the two populations. The performance of Tr-EGT is tested by a numerical simulation experiment and compared with the other multitask optimization algorithms on two practical problems of PPCFP. For future work of the PPCFP, we are interested in considering a constrained a dynamic multiobjective optimization model under dynamic environment such as a change in equipment capacity. In addition, other optimization objectives such as energy consumption can also reflect the production process in more ways than one. In the field of complex engineering manufacturing, which often involves computationally expensive optimization, we will put more effort to study the algorithm shorten time-consuming manufacturing process. Although the proposed multitasking optimization algorithm works well to achieve a good performance on handling most MOPs, there is still negative knowledge transfer, which leads to poor multitasking performance. Therefore, in the future, we would like to investigate how to reduce negative transfer across tasks.

**Author Contributions:** L.J.: conceptualization, formal analysis, investigation, methodology, writing—original draft. Z.Z.: formal analysis, resources, supervision, writing—original draft. K.L.: formal analysis, investigation, methodology, writing—original draft. G.Z.: formal analysis, data curation, software, validation, writing—review & editing. Q.L.: formal analysis, funding acquisition, project administration, resources, writing—review & editing. B.Y.: formal analysis, methodology, resources, visualization, writing—review & editing. Y.F.: formal analysis, methodology, resources, visualization, writing—review & editing. All authors have read and agreed to the published version of the manuscript.

**Funding:** This research has been supported by the annual output of 14,000 tons of high-performance carbon fiber and supporting raw silk construction project of Zhongfu Shenying Carbon Fiber Xining Co., Ltd. under Grant No. 2021-1635-01, the National Natural Science Foundation of China under Grant No. 52075402, No. 52005376.

**Institutional Review Board Statement:** Not applicable.

**Informed Consent Statement:** Not applicable.

**Data Availability Statement:** The datasets obtained during the current work are available from the corresponding author upon request.

**Acknowledgments:** The authors wish to thank the editor and the anonymous referees for their comments which have helped to improve this paper.

**Conflicts of Interest:** The authors declare no competing interest.

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
