# Peer review of "Applying Evolutionary Multitasking for Process Parameter Optimization in Polymerization Process of Carbon Fiber Production"

_applsci, doi:10.3390/app12189378_

Round 1
Reviewer 1 Report
The work presents a multiobjective model for the optimization of the multitask polymerization process of carbon fiber production. Ten different production conditions of the polymerization process are examined. The work is well organized and provides useful information for researchers working in the field. Revision of the following aspects is requested:
- Writing must be revised to correct some grammatical errors. For example, in the abstract “has conducted” must be “has been conducted”, on page 2 (line 42) “can be regard” must be “can be regarded”, on page 8 (line 193) “an good” must be “a good”. Other typos must be checked out and corrected.
- The authors should discuss the effect of introducing dimensionless variables for optimization, so more general conclusions for the management of multitasks can be extracted, e.g., based on characteristic times.
- The authors should include a comparative table of computational times between different algorithms and summarize pros and cons.
Reviewer 2 Report
The manuscript, entitled “Applying Evolutionary Multitasking for Process Parameter Optimization in Polymerization Process of Carbon Fiber Production” by Jin et al., presents an approach to optimize multiple processes to optimize the production of carbon fiber. It is an interesting study, well motivated and clearly presented. However, the authors need to clarify some contents, more specifically, the clarification of the problem statement, before the manuscript gets accepted by MDPI Applied Sciences.
Major points:
· The authors briefly mentioned the carbon black production process, including polymerization, polymer spinning, PAN thermal stabilization, and chemical reaction, etc (Line 32). I would recommend the authors lay out a schematic showing these steps and specify which step this manuscript will discuss. Currently, this manuscript is confusing because it is not clear how the process in this manuscript fits into the complete carbon black production processes.
· The author mentioned that the high-quality PAN is made from AN, copolymer, additives, and solvent (Line 127). What is the copolymer? I think the authors need to clarify what is exactly happening in the polymerizer. It would be nice to have a schematic showing the chemistry of the production (probably in Supporting information). This current manuscript is very vague in describing the exact chemistry.
· In equation 1-8, the authors lay out the equations for optimization, I think the authors need to cite literature properly, mentioning the polymerization mechanism, rather than just giving the equations to the readers without too much explanation. The authors don’t have to explain every parameter, which could be overwhelming. However, the authors can add a few more references here in Page 4-6 in case the readers are curious about the details.
· Could the authors comment on how well the parameter choices represent the actual polymerization process? For examples, the polymerization temperature and time?
Minor Points:
· Please proof read and correct typos. For examples, line 77, “popuplations” should be “populations”. Line 79, “a effective” should be “an effective”. There are more examples like these.
Reviewer 3 Report
This article presented a mechanism model is established for the co-optimization of the polymerization process of carbon fiber production. They also showed various algorithms to support this study. Some minor comments.
In the abstract, please add brief of the results.
It is better to add the references for the Equations.
In Figures 3 and 4, better to make the fonts larger.
Reviewer 4 Report
The paper suggests a multitasking optimization technique based on Transfer learning-based EGT for carbon fibre production.
The Paper is well written and the results are well presented, comparing the performance of the suggested optimization technique with well-known algorithms NSGA-II and MO-MFEA.
Minor comments
1. There exist hard sentences that should be rephrased, such as in 128 : "They are fully mixed at a certain temperature and stirrer stirring" and in 132 "During the reaction process, the rotating speed of the stirrer should be 132 strictly controlled to ensure the molecular weight of the polymer to be controlled within a certain rang"
2. Figure 5 why does it shows only the worst? the average performance seems more fitting in this case.
3. The author should mention the advances in optimization algorithms with search space reduction such as "Improved binary particle swarm optimization for feature selection with new initialization and search space reduction strategies" and " YUKI Algorithm and POD-RBF for Elastostatic and dynamic crack identification.
Reviewer 5 Report
Title : Ok
Abstract : Which method gives you the best results from the 10 proposed
Keywords: you can add more
Comments:
- put each reference near to their specifique property
- results need more discussions
- compare your results with literature ones
- Felow references template and update them (2022)
with regards
-
Round 2
Reviewer 2 Report
The authors presented an improved version of the manuscript with clearer explanations on the process, parameters, etc. The authors clearly highlighted the extensive editing work to make the manuscript a better one. I appreciate the authors' diligence. With the changes, I think the scientific soundness and quality of presentation are significantly improved.